# Horses for Courses in the Era of CARs: Advancing CAR T and CAR NK Cell Therapies

**DOI:** 10.3390/jpm11111182

**Published:** 2021-11-11

**Authors:** Sergey Kulemzin, Igor Evsyukov, Tatiana Belovezhets, Alexander Taranin, Andrey Gorchakov

**Affiliations:** 1Almazov National Medical Research Centre, 197341 Saint Petersburg, Russia; skulemzin@mcb.nsc.ru; 2ITMO University, 197101 Saint Petersburg, Russia; igor.v.evsyukov@mail.ru; 3Institute of Molecular and Cellular Biology, 630090 Novosibirsk, Russia; belovezhec@mcb.nsc.ru (T.B.); taranin@mcb.nsc.ru (A.T.)

**Keywords:** CAR T cell, CAR NK cell, cancer immunotherapy

## Abstract

The adoptive transfer of allogeneic CAR NK cells holds great promise as an anticancer modality due to the relative ease of manufacturing and genetic modification of NK cells, which translates into affordable pricing. Compared to the pronounced efficacy of CAR T cell therapy in the treatment of B cell malignancies, rigorous clinical and preclinical assessment of the antitumor properties of CAR NK cells has been lagging behind. In this brief review, we summarize the biological features of NK cells that may help define the therapeutic niche of CAR NK cells as well as create more potent NK cell-based anticancer products. In addition, we compare T cells and NK cells as the carriers of CARs using the data of single-cell transcriptomic analysis.

## 1. Introduction

Therapy based on the adoptive transfer of T cells genetically modified to express chimeric antigen receptors (CARs) and kill malignant cells has made remarkable strides in clinical applications [1]. The development of approaches to bring down the costs of this technologically complex therapy as well as to apply it to a wider range of human malignancies is an area of active ongoing research [2]. The use of NK cells rather than conventional T cells as allogeneic (universal) CAR carriers represents an attractive alternative, as this cell type is known to conveniently mediate potent antitumor effects in the absence of GvHD [3]. This in turn broadens the selection of donors and cell types for CAR NK cell therapy and it is envisaged to transform the field of immunotherapies employing the adoptive cellular transfer of genetically engineered lymphocytes [4,5,6]. In this review, we discuss the features of CAR NK cells that establish them as a potential rival to significantly more popular CAR T cells in the context of anticancer therapy. We also focus on the challenges that need to be overcome to advance CAR NK cells to the forefront of immunotherapy for human disease.

## 2. Lymphocyte Heterogeneity

T lymphocytes are known to be genetically and phenotypically diverse and comprise several cell subpopulations. Since CAR T cell production typically uses bulk circulating CD3+ cells from peripheral blood as an input, multiple T cell subpopulations become transgenic and express a CAR. A fixed 1:1 ratio of CD4 to CD8 cells has been established to result in improved proliferation and therapeutic activity of CAR T cell products clinically [7,8,9,10]; however, a side-by-side comparison of the typical CAR T cell products and formulations with a defined CD4:CD8 composition in patients is still not available. Similarly, the superior in vivo activity of central and stem central memory-enriched CAR T cell products is now broadly recognized [10,11]. Nonetheless, the possible contribution of minor T cell subsets to the efficacy of CAR T cell therapy merits further analysis [12,13].

NK cells, in turn, are far more heterogeneous as a population. Based on the analysis of the differential expression of surface receptors, thousands of distinct NK cell subpopulations can be identified in the body [14]. It is presently unclear as to whether such immense diversity contributes to the efficacy of CAR NK cell products. However, in theory, it should be a positive factor. Assuming that distinct NK cell subpopulations have different functional properties, some of them may become advantageous for a specific tumor and patient. Such NK cells may be imagined to form a core of the CAR NK cell product. Their preferential expansion may naturally shift the population profile toward a composition showing maximal efficacy. By analogy, we refer to the recently described case of a cancer patient whose CD8+ CAR T cell population at the peak of in vivo activity was derived from a single expanded CAR T cell clone mutant for TET2, accounting for ~90% of this patient’s CAR T cells in the blood [15]. It is possible, then, that a subfraction of CAR NK cells with a favorable expression profile attuned for efficient cancer cell recognition and destruction may preferentially expand in the patient and mediate stronger antitumor activity. Notably, the initial heterogeneity of such a cell population may serve as a positive factor, given that it increases the chances for the presence of a “strong” CAR NK cell clone. It would be interesting to perform a detailed study of the phenotype of persisting CAR NK cell subpopulations in patients with complete responses following CAR NK cell therapy.

## 3. Cytokines

T- and NK-lymphocytes are known to secrete dozens of cytokines that modulate the activity of various cells in the body. Once they are in the patient, CAR T cells may expand a thousandfold and, therefore, the cytokines they produce may have a profound effect on the efficacy and safety of the therapy. In this regard, cytokine cascades that are launched by CAR T cells and that may lead to the development of CRS and neurotoxicity are of particular importance. Such systemic toxicities not only present a threat to the patient’s health, but they also significantly increase the costs of the therapy. It has recently been demonstrated that the key cytokines in the CRS-associated cocktail are produced by myeloid cells rather than by CAR T cells [16,17], with the nature of the triggering cytokine that induces the earliest steps in CRS still remaining enigmatic. It was proposed that the GM-CSF that is secreted by CAR T cells may serve such a role [18], but the exact details of the mechanism on how this may work in a murine xenotransplant setting, given the lack of cross-species reactivity of GM-CSF [17], await further clarification. A possibility remains that there is more than one single trigger for CRS that needs to be further explored in clinical trials.

Using the data from publicly available datasets (https://singlecell.broadinstitute.org/, accessed on 30 October 2021), we performed in silico cytokine gene expression profiling across the major subpopulations of T (CD4+ or CD8+) and NK cells from the peripheral blood of healthy donors (Figure 1). In general, the patterns of cytokine gene expression are surprisingly similar for CD8+ T cells and NK cells. The latter are characterized by a slightly higher expression of IFN-g and decreased expression of IL-32. Furthermore, relative IL-16 expression is more pronounced in NK cells than in CD8+ T cells.

Unfortunately, this type of bioinformatic analysis does not discriminate between the different isoforms of IL-32, although their activities are known to be significantly different: IL-32gamma has a clear pro-inflammatory function, whereas this is not the case for the major isoform (IL-32alpha) [19]. Therefore, the comparative analysis of different IL-32 isoforms in CAR T and CAR NK cell products appears to be useful.

## 4. Receptors

It is clear that the strongest signaling in CAR T/NK cells is launched by the CAR itself. Nonetheless, there are multiple other receptors, both activating and inhibitory, whose engagement may profoundly alter the activity of a CAR-expressing lymphocyte.

### 4.1. Inhibitory Receptors

The negative effects of PD-1 signaling on CAR T cells are well documented [20,21], and clinical trials of CAR T cells that are genetically engineered to withstand PD-1 signaling or that have the Pdcd1 gene knocked out are under way (NCT04850560, NCT03706326, NCT03525782, NCT03545815, NCT03747965, NCT03208556, NCT02937844, etc.). Our analysis of the expression data did not reveal high PD-1 expression in either T or NK cells. However, it is noteworthy that the cells used for analysis were obtained from the peripheral blood of healthy donors, which is consistent with the idea that the PD-1 expression in pathological conditions may well be different.

It has been reported that multiple myeloma (MM) patients have NK cells expressing PD-1 and their exposure to autologous PD-L1-expressing MM cells leads to the downregulation of NK cell activity [22]. This may not be relevant for the CAR NK cell products, however, as allogeneic NK cells lacking PD-1 expression, rather than the autologous ones, are typically considered as the cellular source for manufacturing.

When compared to T cells, NK cells are clearly distinguished by the increased expression of a wide range of inhibitory receptors (Figure 2). Along with KIRs, KLRB1, LAIR1, Siglec7, CD300A, and TIM-3 (HAVCR2) could be detected. As for the latter, TIM-3 is a marker of mature NK cells. Notably, such cells are fully competent in terms of cytokine production and cytotoxic activity [23].

Inhibitory receptor Siglec7 interacts with sialoglycosides, which are frequently overrepresented on the surface of tumor cells [24]. Accordingly, such tumor cells inhibit the cytotoxic activity of NK cells by cross-linking Siglec7. Notably, CMV-specific NK cells have reduced Siglec7 expression [25]. Recently, K562 cells (which are often used for the expansion of NK cells in vitro) were shown to express the ligand for Siglec7, which is a sialylated form of CD43 [26]. Therefore, it is quite likely that the interaction between CD43 and Siglec7 could reduce the efficiency of NK cell expansion on such K562-derived feeders.

Collagen is one of the ligands of another inhibitory receptor LAIR 1. Increased collagen synthesis by a number of tumors may, therefore, negatively affect the cytotoxic activity of NK cells by inducing the signals from LAIR1 [27]. A similar mechanism of downregulation of NK cell activity through the stimulation of their receptor CD300A by its ligand phosphatidylserine has also been implicated in tumor resistance to NK lysis [28].

The expression of KLRB1 begins early in NK cell differentiation and provides activating, rather than inhibitory, signaling to immature NK cells [29]. In the context of mature NK cells, however, KLRB1 shows pronounced inhibitory activity [30]. Furthermore, blocking the interaction between KLRB1 and its ligand LLT1 results in the stronger cytotoxicity of NK cells against triple-negative breast cancer cells [31]. Moreover, prostate cancer cells have been shown to actively exploit the LLT1/KLRB1 axis and avoid destruction by overexpressing LLT1 [32]. Therefore, of the inhibitory receptors listed hereinabove, KLRB1 appears as a promising target for gene disruption in CAR NK cells, particularly if such cells are intended to treat LLT1-overexpressing tumors.

In addition to the expression of the ligands of inhibitory receptors, tumors employ a broad range of strategies to blunt the functionality of adoptively transferred immune cells and shift the balance toward immunosuppressive/tolerance pathways. Collectively referred to as the tumor microenvironment (TME), these factors include local production of various cytokines (e.g., TGFbeta), enzymes (CD39, CD73) and small molecule metabolites (PGE2, adenosine, NO, K+) that cumulatively inhibit NK/T cell activity [33,34,35]. Furthermore, tumor cells and stroma are known to compete with intratumoral immune cells for nutrients and oxygen [36] (see below). Understanding which factors predominate in the context of which tumors may become instrumental for devising the approaches to modulate metabolic reprogramming of CAR NK/T cells and enhance their anticancer activity, particularly in solid tumors. Several notable examples of when such approaches were productive for CAR T cells or NK cells (genetic or pharmacological targeting of TGFbetaRII, A2AR, etc. [37,38,39]) are expected to be equally potent in the context of CAR NK cells [40] and warrant further investigation.

### 4.2. Activating Receptors

Unlike T cells, where the dominant activating mechanism is orchestrated by the TCR, NK cells lack a similar “hub” receptor. Therefore, the overexpression of activating receptors or combinations thereof in NK cells to stimulate their antitumor activity appears highly promising (Figure 2).

Only a handful of co-receptors, such as CD5, CD27, and CD28, are expressed by T cells and are virtually absent from NK cells. Of interest is that, despite the lack of CD28 expression in resting NK cells, CD28-based second-generation CARs display stronger activity in NK cells than the first-generation CARs [41,42]. This may be attributed to the fact that the major elements of the signaling machinery are present in NK cells at levels comparable to those of T cells (Figure 2). Surprisingly, NK cells express more CD3zeta (CD247) than T cells. Therefore, a side-by-side comparison of the cytotoxic activity induced by the first-generation CAR in T and NK cells would be interesting, since a higher dose of CD3zeta potentially could initiate stronger signaling in NK cells.

It is noteworthy that NK cells are characterized by a very high expression of DAP12 (TYROBP), an adapter molecule less typical for T cells. In this regard, it seems perfectly reasonable to use DAP12 as a signaling motif when tailoring CARs for NK carriers. Nevertheless, to date, the use of the DAP12 sequences in CAR structure remains rather exotic [43,44,45].

Accordingly, a comprehensive study of various signaling motifs within CARs could outline the “ideal” intracellular part of CARs for NK cells or would show the need to optimize signaling for each specific scFv on a case-by-case basis, depending on the exact architecture of the CAR (scFv, hinge, TM).

## 5. Immune Synapse and Cytotoxicity

Immune synapse and cytolysis are mechanistically quite similar for T and NK cells, yet several differences also stand out [46]. First, lytic granules in the T cells are formed after the cell has acquired an effector phenotype, whereas such granules are pre-formed in NK cells and are present even in resting cells [47]. Second, integrin LFA-1 (ITGAL) and yet another integrin, MAC-1 (ITGAM), are expressed significantly higher in NK cells compared to T cells (Figure 3). The surface expression of these two molecules is key in the initial steps of synapse formation, and it is likely for this reason that CAR NK cells form more stable pre-synapses than CAR T cells. There is significant homology between the processes occurring downstream of the activating receptor crosslinking in T and NK cells, and this is indirectly supported by similar expression levels of the genes involved, CDC42, VAV1, ZAP70, LAT, SLP-76 (LCP2), PYK2 (PTK2B), and CIP4 (TRIP10) (Figure 3). It follows from the published data and our transcriptome profiling that NK cells express granulysin, perforin as well as several granzymes at higher levels than CD8+ T cells, not to mention CD4+ T cells [48] (Figure 3). To our knowledge, no side-by-side comparisons of CAR T vs. CAR NK cell cytotoxicity have been performed. We expect that the higher expression of cytolytic molecules by NK cells may actually translate into their superior cell-killing activity. Of particular interest is the upregulated expression of granulysin by NK cells. Cancer cells have been reported to become resistant to granzyme B-mediated destruction [49]. Therefore, the combined activity of granzymes and granulysin produced by NK cells may lead to stronger cytolysis.

It must be emphasized that the very necessity of a “classic” mature immune synapse for the destruction of the target cell has long been questioned [50]. Multiple cell killings of several target cells by T cells were observed, with the “classical” synapse being formed with one cell and lytic granules being directed to a different cell [51]. Furthermore, it has recently been found that CAR T cells can form a non-classic synapse with an overall shorter signaling and dissociation time, which may also be uncoupled from LFA-1 clustering [52]. In all likelihood, CARs having distinct antigen binding and signaling structures will form distinct types of synapses. Therefore, the biology of both CAR T and CAR NK-induced immune synapses is an area of exciting future research, and the extensive optimization of CAR design may help obtain cell products with superior antitumor properties.

The second, somewhat slower mechanism of tumor cell killing relies on the signaling downstream of the Fas receptor upon the engagement of FasL (FASLG) that is displayed by the effector cell. FasL molecules are constantly present on the surface of T/NK cells. However, most of the FasL payload is delivered to the target cell by lytic granules [53]. Compared to T cells, a fraction of NK cells expresses more FasL (Figure 3), which may indicate that there are NK cell subpopulations potently inducing perforin/granzyme-independent target cell lysis. TRAIL (TNFSF10) is one more ligand of pro-apoptotic receptors that can efficiently lyse the tumor cells [54]. The expression levels of TRAIL are equally low across T and NK cells.

## 6. Hypoxia

One of the obstacles for the efficient antitumor activity of CAR T/NK cells is hypoxia. To our knowledge, no functional comparisons of CAR T vs. CAR NK cells in hypoxic conditions have been performed. Yet, NK cells appear to be significantly more sensitive to hypoxia, as they display slower proliferation and impaired cytotoxicity under these conditions [55]. No obvious expression differences between T and NK cells were observed for the genes encoding oxygen sensors (PHD1, PHD2, PHD3, FIH) (Figure 3).

Therefore, it remains to be explored why T and NK cells are so profoundly different in terms of their response to hypoxia. Two approaches to mitigate the detrimental effects of oxygen deprivation on NK cells were proposed and include pre-activation with IL-2 [56] and the downregulation of miR-210 levels [55].

## 7. Homing and Taxis

The particular localization of cellular effectors in the body of the patient is extremely important, since no matter how active the effector cell is, if it is not localized in the tumor focus, it will not be able to have an antitumor effect. T and NK cells differ most significantly in the expression of CX3CR1 (Figure 4). This chemokine receptor is expressed on a significant proportion of NK cells but is not typical for T lymphocytes. CX3CR1 is expressed on mature activated NK cells, characterized by a high level of cytotoxicity and the ability to migrate into tissues, including the central nervous system (although some of the data on migration were obtained in mouse models) [57,58]. It has been shown that the ectopic expression of CX3CR1 on T cells improves their trafficking to tumors expressing CX3CL1, a natural ligand for CX3CR1 [59]. From this perspective, NK cells appear more attractive since they naturally express this receptor without the need for genetic manipulation. However, the careful analysis of the percentage of CX3CR1+ cells across various CAR NK cell preparations is required, as there are reports indicating that the most easily transduced subpopulation of NK cells typically lacks this receptor [60].

A fraction of NK cells is characterized by their higher expression of CXCR2 (Figure 4). However, in a vast majority of NK cells, it still remains at a low level. It has been shown that the genetic modification of NK cells to increase the expression of CXCR2 improves their trafficking to renal cell carcinoma cells [61]. Likewise, in mouse models, T cells modified to express CXCR2 reach melanoma cells better than unmodified T cells [62]. A similar effect has been observed for ovarian or pancreatic tumor xenografts [63].

Another chemokine receptor, CXCR3, is expressed by both T and NK cells at a moderate level, and it has been shown that the NK cells found in anaplastic thyroid cancer foci express CXCR3 at a higher level than those found in the peripheral blood NK cells, which indicates the involvement of this receptor in the redirection of NK cells to the tumor focus [64]. For T cells, CXCR3 has been shown to be critical for efficient extravasation and trafficking to the tumors [65].

Neither T nor NK cells express CCR2 receptors at a high level. Meanwhile, the ectopic expression of this protein increases the trafficking of T cells to mesothelioma foci [66]. For myeloma therapy, it seems that a decrease in the expression of CXCR3 by NK cells will improve therapy [67]. In contrast, to achieve the better migration of T cells to Hodgkin lymphoma sites, CCR4 expression is essential [68]. To summarize, different types of cancer require the fine-tuning of the chemokine receptor expression on CAR T/NK cells.

## 8. Manufacturing of CAR T and CAR NK Products

Hundreds of launched clinical trials and several registered CAR T cell products have required significant efforts to develop and optimize the industrial production of CAR T cells. The introduction of CAR NK cells in clinical practice has just begun, so CAR NK cell manufacturing has a lot of room for improvement. What are the similarities and differences in the manufacturing processes of these two types of cellular products? First of all, unlike CAR T, CAR NK cells can be used as an allogeneic product without additional modifications. In this regard, the most frequent raw materials for the production process are not the patient’s own NK cells but rather umbilical cord blood [3] or the peripheral blood of a haploidentical (NCT04887012, [69]) or HLA-mismatched donor [69]. Furthermore, advancing the platforms based on iPSC-derived CAR NK cells may help address the need for large-scale manufacturing and on-demand multiple dosing of highly homogeneous and standardized cell products [70,71]. Primary NK cells are notoriously difficult to transduce with lentiviruses [72], but this problem can be solved by the appropriate pseudotyping and retroviral delivery [73,74]. It also seems promising to use the non-viral modification of NK cells, e.g., transposons [75,76]. Another difference in manufacturing processes is the need to use feeder cells for the efficient expansion of CAR NK cells. Most often, K562 cells expressing cytokines and ligands of activating receptors [74,77] on their surface are used for this purpose. Before mixing with NK cells, the feeder cells are irradiated. However, their introduction into the production cycle requires additional quality control steps in order to exclude contamination with feeder cells and components. At the same time, more than 10,000-fold expansion of NK cells was reported for a feeder-based protocol [78], thereby making it possible to produce enough CAR NK cells for multiple patients in one production cycle. Therefore, despite the increased complexity of the production of one batch of CAR NK cells, the per patient cost of the product can be lower, since the cost of quality controls will be split between several doses of the product.

Another possible difficulty on the route to widespread use of CAR NK cells is the increased sensitivity of NK cells to cryoconservation. There is conflicting information about the functionality of NK cells after thawing. Some groups do not observe any critical decrease in viability and functionality [79,80], whereas others have reported a significant decrease in cytotoxicity after thawing [81]. It is noteworthy that, in a recent clinical trial of CAR NK cells for the therapy of CD19-positive cancers, CAR NK cell preparation was administered to patients fresh [3].

Recently, Weber et al. demonstrated that the transient inhibition of CAR signaling may result in enhanced/restored CAR-T cell functionality by preventing or reversing cell exhaustion [82]. Whether or not continued stimulation and cytokine exposure implemented under current CAR NK manufacturing protocols may similarly drive these cells to the acquisition of suboptimal phenotypes is currently not known. Therefore, adjusting CAR NK cell cultivation conditions and the inclusion of “rest” periods may similarly be productive and warrants further analysis.

## 9. Conclusions

Despite their significant similarities as cytotoxic carriers of CARs, T and NK cells are rather different (Figure 5). NK cells contain more granzymes, which most likely makes them more cytotoxic. The abundance of activating receptors that recognize stress-induced ligands allows one to expect that the CAR NK cells can effectively lyse the tumor even in the absence of a specific marker recognized by the CAR, which should translate into a reduced risk of tumor escape. At the same time, there is still ample space for modifications and improvements to the CAR NK design. Besides tailoring the CAR structure, homing and taxis could be corrected, the expression of inhibitory receptors can be edited to increase resistance to the suppressive environment, and survival under hypoxic conditions could be enhanced.

Most likely, the question “Which cell type–T or NK—is the best carrier for a CAR?” is wrong, or at the very least should be placed in the context of the specific malignancy. Furthermore, the history of cell therapy shows that no in vitro experiments or mouse model studies can reliably predict its efficacy in real patients. That is why the data obtained in the clinical trials of CAR NK cells are extremely important. So far, encouraging results are available from one such trial [3]. Intriguingly, recent reports have highlighted an opportunity to combine CAR T and NK cells (expressing CAR or not) in one product, with the advantage of reduced negative side effects of CAR T cells and enhanced antitumor activity [83,84]. Hopefully, the coming years will bring us an optimal strategy for using CAR T/NK cells in the treatment of cancer.

## Figures and Tables

**Figure 1 jpm-11-01182-f001:**
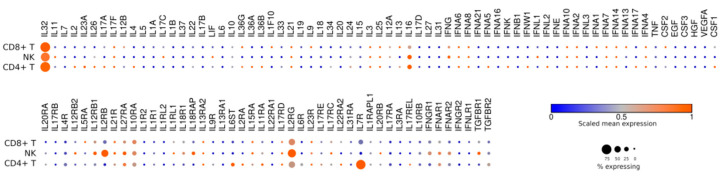
Expression plot for cytokine receptors and cytokine-encoding genes across the major subpopulations of human T and NK cells.

**Figure 2 jpm-11-01182-f002:**
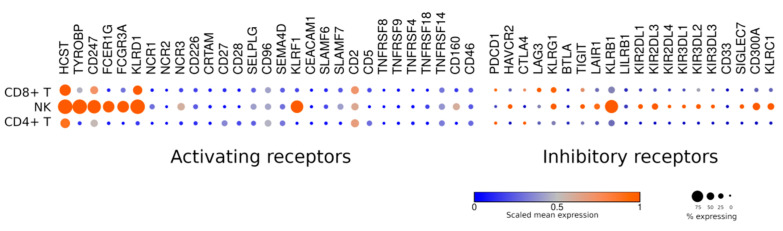
NK cells express a wide variety of activating and inhibitory receptors.

**Figure 3 jpm-11-01182-f003:**
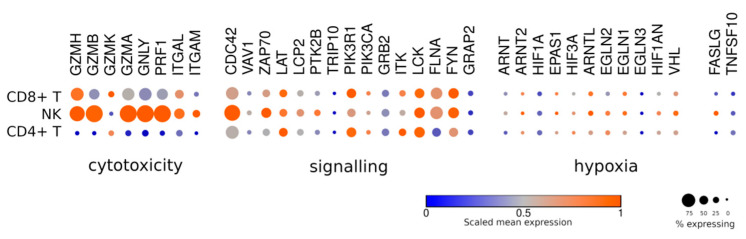
Expression plot for different groups of genes across the major subpopulations of human T and NK cells. “Cytotoxicity” genes encoding perforin and granzymes and the synapse-related proteins ITGAL and ITGAM; “signaling” genes encoding activation-related proteins; “hypoxia” genes involved in the cell response to hypoxia.

**Figure 4 jpm-11-01182-f004:**
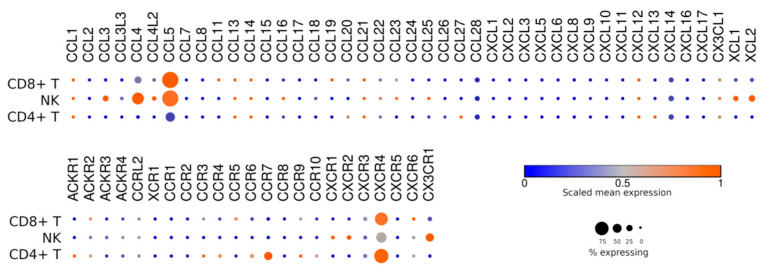
Homing of NK cells as well as T cells could be extensively improved.

**Figure 5 jpm-11-01182-f005:**
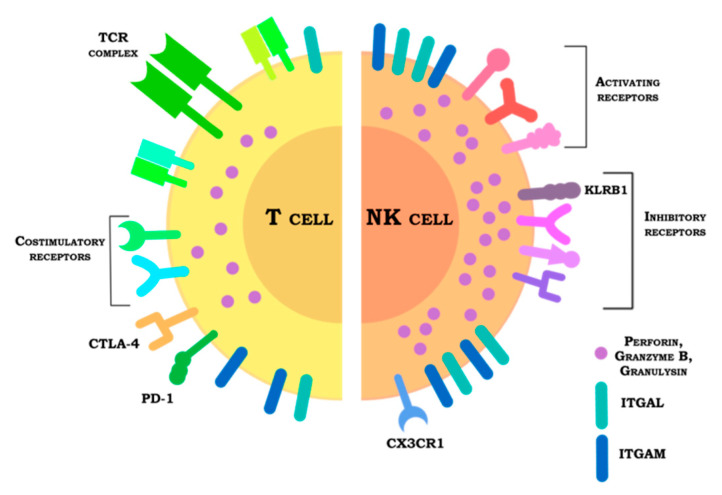
Schematic comparison of T and NK cells in the context of their use as CAR carriers. Unlike T cells, NK cells do not have TCR, but rather express a much greater variety of both activating and inhibitory receptors. The expression patterns of the inhibitory receptors differ between these cell types: in case of T cells, clinically relevant is expression of CTLA-4 and PD-1 receptors, for NK cells the potential checkpoint is KLRB1 receptor. NK cells express more perforin and granzymes and have a higher density of ITGAL and ITGAM on their surface, potentially making them the best cancer cell killers. For both T and NK cells, it is reasonable to express additional chemokine receptors to modulate taxis.

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
