# Peer review of "Horses for Courses in the Era of CARs: Advancing CAR T and CAR NK Cell Therapies"

_jpm, 2021, doi:10.3390/jpm11111182_

Round 1

Reviewer 1 Report

In this manuscript,  authors have reviewed the CAR-NK cells and their application in cancer immunotherapy. Authors have compared the potential advantages and pitfalls of NK-cell therapy to CAR-T cell therapy. The manuscript is well written with detailed discussion on heterogeneity of these cells, presence of inhibitory and activating receptors, cytokine release, and few other factors leading to efficiency of these therapies. 

The manuscript should be accepted with following additions:

  1. Authors have mentioned ease of manufacturing and genetic modification of NK cells in abstract but it is missing from the main body of the paper.
  2. A discussion on scalability of the production should be included in the manuscript as well. 
  3. A more detailed conclusion with focus on why assessment of clinically relevant studies with NK cells is lagging would improve the review even further.  

Author Response

Dear Reviewer,

Thank you for your time and consideration!

We modified text of our manuscript to meet your suggestions. Please find our detailed response:

> 1. Authors have mentioned ease of manufacturing and genetic modification of NK cells in abstract but it is missing from the main body of the paper.

> 2. A discussion on scalability of the production should be included in the manuscript as well. 

We included additional section “Manufacturing of CAR T and CAR NK products”, which covers modification and expansion of CAR NK cells.

> 3. A more detailed conclusion with focus on why assessment of clinically relevant studies with NK cells is lagging would improve the review even further.

Conclusion is now modified.

Reviewer 2 Report

Dear Authors,

The review is well written and it is informative, Can you add a section on the CAR-T/NK-T therapy and Tumor immunosuppression because it is a major factor for the all the immunotherapy.

Author Response

Dear Reviewer,

Thank you for your time and consideration!

We included extended discussion of CAR NK cells interaction with TME in the end of “inhibitory receptors” section.

To improve quality of English we will send manuscript to professional proofer.